# Passive Brain–Computer Interface Using Textile-Based Electroencephalography

**DOI:** 10.3390/s25196080

**Published:** 2025-10-02

**Authors:** Alec Anzalone, Emily Acampora, Careesa Liu, Sujoy Ghosh Hajra

**Affiliations:** Department of Biomedical Engineering and Science, Florida Institute of Technology, Melbourne, FL 32901, USA; aanzalone2021@my.fit.edu (A.A.);

**Keywords:** passive brain-computer interface (pBCI), electroencephalography (EEG), textile electrode, cognitive state classification, support vector machine (SVM)

## Abstract

**Background:** Passive brain–computer interface (pBCI) systems use a combination of electroencephalography (EEG) and machine learning (ML) to evaluate a user’s cognitive and physiological state, with increasing applications in both clinical and non-clinical scenarios. pBCI systems have been limited by their traditional reliance on sensor technologies that cannot easily be integrated into non-laboratory settings where pBCIs are most needed. Advances in textile-electrode-based EEG show promise in overcoming the operational limitations; however, no study has demonstrated their use in pBCIs. This study presents the first application of fully textile-based EEG for pBCIs in differentiating cognitive states. **Methods:** Cognitive state comparisons between eyes-open (EO) and eyes-closed (EC) conditions were conducted using publicly available data for both novel textile and traditional dry-electrode EEG. EO vs. EC differences across both EEG sensor technologies were assessed in delta, theta, alpha, and beta EEG power bands, followed by the application of a Support Vector Machine (SVM) classifier. The SVM was applied to each EEG system separately and in a combined setting, where the classifier was trained on dry EEG data and tested on textile EEG data. **Results:** The textile EEG system accurately captured the characteristic increase in alpha power from EO to EC (*p* < 0.01), but power values were lower than those of dry EEG across all frequency bands. Classification accuracies for the standalone dry and textile systems were 96% and 92%, respectively. The cross-sensor generalizability assessment resulted in a 91% classification accuracy. **Conclusions:** This study presents the first use of textile-based EEG for pBCI applications. Our results indicate that textile-based EEG can reliably capture changes in EEG power bands between EO and EC, and that a pBCI system utilizing non-traditional textile electrodes is both accurate and generalizable.

## 1. Introduction

Passive brain–computer interfaces (pBCIs) combine electroencephalography (EEG) and machine learning (ML) to enrich human–machine interactions through devices and applications that adapt to the human mental state [1]. Traditional brain–computer interfaces (BCIs) utilize brain measurements to affect some change in the world (e.g., prosthetics); more generally, a BCI system requires a brain imaging modality, typically electroencephalography (EEG), to measure activity and a computer system to interpret that activity and output the user’s intended effect. Unlike traditional BCIs, which rely on explicit user input, pBCIs rely on the user’s implicit or passive inputs [2], which are an interpretation of the user’s cognitive state (e.g., eyes closed, fatigue, stress, and workload), commonly referred to as cognitive state monitoring. Determining a cognitive state requires the quantification of EEG data, which exhibits a high degree of variability [3]. Machine learning (ML), a class of statistical methods designed to recognize patterns in data, has shown promise in the interpretation of physiological data. ML classifiers, including Support Vector Machine (SVM), Gaussian Process (GP), and K-Nearest Neighbor (KNN), have been applied to EEG measurements, and can be used to monitor the spontaneous neural dynamics relating to changes in cognitive states [4,5,6,7].

Current applications focus on areas where the classification of operator states can improve safety or performance, especially in the complex environments experienced by airplane or helicopter pilots [8]. Previous studies have investigated pBCIs in several scenarios, including virtual reality flight training [9], simulated anti-collision radar monitoring [10], and simulated drone piloting [11]. The latter took a different approach by predicting performance instead of mental state, highlighting the value of outcome-focused classifications in pBCIs [11]. In [10], researchers combined a standard BCI and a passive BCI system based on minimal dry EEG to achieve a closed-loop system. Prior work has focused on simulating non-laboratory settings; however, the systems developed lack practicality for deployment in the target environments. In [9,11], researchers employed a standard research-grade EEG system, which typically samples at 500–1000 Hz and employs active wet electrodes or dry metal electrodes with 64 to 256 channels. Wires extend 1 to 2 m from the device and lead to large amplification, filtering, and digitization hardware, which must also be connected to the PC used to record the measurements. Although such systems offer high precision and stability, their cumbersome setup (e.g., bulky hardware and long wires) limits their practical deployment in non-laboratory scenarios. Additionally, the metal electrodes used by research-grade wet and dry EEG systems are incompatible with many real-world settings where cognitive state monitoring using pBCIs has shown promise, including aviation, construction, sports, space, and the military where the use of helmets renders any system with metal electrodes unusable due to safety concerns [3,12,13].

Recent advances in electrode technology may provide a solution. These include flexible dry electrodes, semi-dry electrodes, 3D-printed flexible electrodes, and textile-based electrodes [14]. Among these, textile electrodes show the most promise for overcoming the operational limitations of standard metal electrodes. Constructed using an insulating fiber base and conductive fibers to carry the signal, they do not require gel and can be integrated with almost any type of fabric to house the system. The most common is a headband-style garment, which allows for reliable contact with frontal, temporal, and occipital regions. Textile-based electrodes have been utilized in neonatal applications, intensive care units, electrical stimulation, electromyography (EMG), and electrocardiography (ECG) [15,16,17,18]. A pBCI is commonly stated to be a potential application of textile electrodes, but very few studies have actually demonstrated this. Of those, it is common to see the devices used referred to as textile- or garment-based; however, these still utilize metal electrodes, which renders them unusable in many of the proposed real-world applications [19]. To our knowledge, no study has evaluated whether fully textile-based EEG systems can be used for pBCI cognitive state monitoring applications.

Many types of physiological monitoring devices are already widely used (e.g., smartphones, smartwatches, and fitness trackers) to monitor physical health parameters, such as heart rate, blood pressure, oxygen saturation, activity level, and sleep. The domain of cognitive state monitoring applies the same concept to the brain, measuring the mental state instead of the physical state [20]. This can range from a simple detection of eyes open (EO) vs. eyes closed (EC) to higher-complexity states such as mental workload, stress, and level of vigilance [14,21,22]. The feasibility of monitoring these complex cognitive states has been demonstrated in realistic operational settings. Page et al. successfully captured cognitive workload differences during pilot-like tasks using the Multi-Attribute Task Battery (MATB) and low-density EEG, showing that EEG-based systems can reliably differentiate between low, medium, and high cognitive load conditions in complex operational scenarios [23]. Their work provides evidence that cognitive state monitoring can move beyond simple binary classifications to capture nuanced cognitive load differences in naturalistic settings. Although such advanced applications are promising, the foundational validation of novel EEG systems—particularly those using unconventional materials such as textiles—still relies on well-established paradigms. Here, we evaluate the cognitive state monitoring ability of a pBCI using fully textile-based EEG with the simple and robust EO vs. EC paradigm, which is widely used due to its reproducible spectral changes. There are well-documented increases in alpha (α) frequency power, as well as changes in beta (β), delta (δ), and theta (θ), when the eyes are closed compared with open. Closing the eyes shows an increase in α power, which indicates the inhibition of visual processing corresponding with the lack of visual input [24]. EO and EC classifications are commonly employed as a method for validating classifiers developed for use in disease detection or BCI [25]. To that end, the present study aimed to examine the feasibility of using a fully textile EEG-based pBCI for the evaluation of cognitive states. We hypothesized that a textile electrode EEG system would be able to capture the change in alpha power between EO and EC, exhibiting comparable performance to a standard dry-electrode EEG, and that a passive BCI system utilizing textile EEG would be able to differentiate between the EO and EC cognitive states.

## 2. Materials and Methods

### 2.1. Participants

EEG recordings were collected from 10 healthy participants aged 27.8 ± 3.7 years, consisting of 5 females and 5 males. The purpose and parameters of the study were fully explained, and each participant signed an informed consent agreement. Participants also received an economic compensation of EUR 25. Ethical approval was obtained from the Institutional Review Board of the Florida Institute of Technology for the secondary analysis of existing human physiological data (Protocol #24-177).

### 2.2. EEG Systems

The textile-based EEG system was entirely constructed of textile materials on a headband platform composed of polyester. Recording locations consisted of the F7, Fp1, Fp2, and F8 EEG channels based on the 10–20 international system. The textile electrodes were constructed from three-strand silver-coated nylon fibers embroidered in 3 directions, with a distance between yarns of 1 mm and linear resistance of 114 Ω/m. The ground electrode was located at Fpz, and the reference was located at the upper union between the left ear and head. Signal transmission from the “textrode” was achieved through layering of conductive materials intended to mimic the active shielding properties of coaxial cables. The main transmission wire was embroidered with eight-strand silver-coated nylon and insulated using vinyl. Finally, a conductive layer made of PA/Nylon 6.6 and coated with Ag, Cu, and Sn was added to shield both sides. The textrode was connected to the transmission wire by embroidering the wire directly onto the fabric, the textile EEG headband is shown in Figure 1.

The dry-electrode EEG system, shown in Figure 2, mirrored the textile-based system, with four recording locations at F7, Fp1, Fp2, and F8, and a ground at Fpz. The reference was located on the participant’s earlobe. Electrodes were standard Ag/AgCl dry electrodes. Both headbands were equipped with the same high-input impedance amplifier (50 GΩ), with a CMRR of 100 dB.

### 2.3. EEG Acquisition and Preprocessing

EEG measurements were acquired from Lopez-Larraz et al. and sampled at 256 Hz [26]. Participants performed the following two tasks: 3 min of resting state with eyes closed and 3 min of resting state with eyes open. The tasks were performed using the two headbands sequentially, with half of the participants starting with textile EEG and the other half starting with dry EEG. The data were filtered from 1 to 50 Hz using a fourth-order Butterworth filter to remove baseline and high-frequency noise and epoched into 2 s segments per subject per channel, resulting in a total of 3600 epochs per EEG system. To address motion and blink artifacts, epochs were removed where the individual channel voltage exceeded ±100 µV. A total of 870 epochs were removed from the dry EEG data, and 979 total epochs were removed from the textile EEG data. This method, outlined by [25], has been compared with a traditional preprocessing pipeline and the results indicate comparable performance for blink and motion artifact removal.

### 2.4. Standard Power Analysis

The power spectra for each signal were extracted using Welch’s power spectral density estimate, with a Hanning window size of 128 samples and 50% overlap [27]. The signal power in the delta (0.5–4 Hz), theta (4–7 Hz), alpha (8–13 Hz), beta (13–30 Hz), and gamma (30–50 Hz) frequency bands was obtained by taking the average from each band of interest. The ability of textile EEG to differentiate cognitive states through a standard power analysis was assessed using a Friedman rank sum test, and differences in each power band were assessed using pairwise Wilcoxon rank sum tests with a Benjamini–Hochberg correction for multiple comparisons. Results were considered to be significant if the *p*-value was less than 0.05. Effect sizes (r) were calculated using the Z-statistic extracted from the Wilcoxon rank sum test divided by the square root of the effect size.

### 2.5. Passive Brain–Computer Interface

The pBCI development consisted of the following three phases: model selection, feature engineering, and optimization. The model selection considered the following 5 ML classifiers of interest: linear kernel Support Vector Machine (SVM), Gaussian kernel SVM, K-Nearest Neighbor (KNN), Random Forest (RF), and logistic regression (LR). Each was selected according to the commonly used classifiers for EO and EC classification in the literature. The models selected were limited to less-complex models to maintain a minimum computational requirement designed to mimic the limitations of real-world applications. The initial feature set consisted of the electrode location of each signal; the mean, maximum, and minimum power of the delta, theta, alpha, beta, and gamma frequency bands; and the skewness, kurtosis, and Shannon entropy of the signal, resulting in 18 total features. Each model was evaluated by randomly selecting 90% of the data to train the classifier and the remaining 10% was used to test; this was repeated 10 times, each with a new classifier instance so that the models were trained and tested on all data within a ten-fold cross-validation [28]. This was performed using both standalone dry and textile data as well as in a combined setting using dry EEG data for training and textile EEG data for testing. Each was assessed for accuracy, precision, recall, F1 score, and fit time for each permutation, and the scores reported were the means of all 10 folds (Table 1). The final combination of dry and textile data mimics what the real-world use of this system might look like, where a standard-type EEG system (e.g., a dry electrode) would be used as a baseline calibration and textile EEG would be deployed for outside-the-lab use.

The linear SVM classifier was selected as it represents the best compromise between high accuracy, precision, recall, F1 scores, and a low fit time. This was used as the base for developing the pBCI system, with three models corresponding with each of the separate and combined datasets. Feature selection was conducted using a maximum relevance minimum redundancy (MRMR) feature-sorting algorithm. The new feature set was applied to the dataset and hyperparameters were tuned for optimal performance using a 5-fold cross-validation and grid-search function from the Python (version 3.13.7) Scikit-Learn package (version 1.6.1) conducted over the C and gamma parameter ranges of [0.1, 1, 10, 100, 1000] and [1, 0.1, 0.01, 0.001, 0.0001]. The final classification results were obtained by applying the same 10-fold cross-validation method used for the classifier selection, with updated feature sets and hyperparameters for each of the three final models.

Statistical validation of the classification results was conducted to assess both the score reliability within each model and differences between models. The statistical significance of within-model classification results was evaluated using non-parametric permutation testing [28]. This involved randomly redistributing target labels and performing the same cross-validation procedure, repeated 1000 times. This resulted in a null distribution, against which the true classification scores were compared. Probabilities less than 0.05 were considered to be significant. A between-model comparison was conducted using the accuracy, precision, recall, F1, and fit time scores for each fold of the cross-validation procedure. These were first assessed for normality using a Shapiro–Wilk test then compared between models using a Kruskal–Wallis rank sum test. A post hoc analysis was conducted using pairwise Wilcoxon rank sum tests.

## 3. Results

Figure 3 shows a comparison between the standard EEG power bands for the dry and textile-based EEG systems. The dry EEG system captured significant differences between conditions in the delta (*p* < 0.0001; r = 0.66), theta (*p* < 0.0001; r = 0.64), and alpha (*p* < 0.01; r = 0.13) power bands. Along with the expected increase in alpha power when the eyes were closed, there was a notably sharp decrease in delta band power with eyes closed. The textile EEG system reproduced the expected increase in alpha power (*p* < 0.01; r = 0.13), confirming its sensitivity to changes in cognitive states. Similar decreases in the delta and theta bands were captured (*p* < 0.0001 and r = 0.66 and *p* < 0.0001 and r = 0.64, respectively), but with much smaller differences in power between the two conditions. In addition, the power values captured from textile EEG were lower overall than those from the dry EEG system, with a maximum value of 16 dB captured in the textile system and a maximum value of 30 dB captured in the dry-electrode system. Figure 4 presents the full-power spectra captured at each electrode compared between each system and condition, indicating that, at an individual channel level, both systems captured similar values across frequencies and that there was a clear differentiation between the two conditions.

Feature selection from the EEG data resulted in a set of 10 features that provided biologically relevant information and resulted in the highest classification scores. These were the delta, theta, and alpha power bands; the maximum and minimum delta and theta power values; and the skewness, kurtosis, and Shannon entropy values. A statistical analysis revealed significant differences between conditions for each feature (Figure 5 and Figure 6). Although MRMR selection was applied to both dry- and textile-electrode data, only dry EEG features were used for model evaluation for both consistency and to examine generalizability.

Cognitive state classification ability was assessed using both standalone systems and combined data. Table 2 shows the classification results, most notably, the standalone dry EEG system resulted in 94% accuracy and an F1 score of 93%. The textile EEG system had a similar accuracy at 94% and an F1 score of 94%. The generalizability assessment, which used the dry-electrode data to train the model and textile-electrode data to test the model, resulted in 92% accuracy and an F1 score of 92%, indicating strong cross-platform performance. Additionally, a between-subject omnibus comparison revealed no significant difference in accuracy (*χ*^2^(2) = 0.36; *p* = 0.83), precision (*χ*^2^(2) = 4.49; *p* = 0.11), or F1 score (*χ*^2^(2) = 0.27; *p* = 0.87). The Kruskal–Wallis test found significance in the recall (*χ*^2^(2) = 9.68; *p* < 0.05); however, the post hoc analysis revealed no significance between individual models (*p* > 0.05).

## 4. Discussion

The objective of this study was to examine the feasibility of using an EEG system employing minimal, non-traditional electrodes for pBCI applications. This was evaluated first through a standard power band analysis and later through cognitive state classification using a novel textile EEG system developed by [26]. The comparison of interest was between the EO and EC cognitive states, which is a well-studied phenomenon commonly used to evaluate and calibrate pBCI systems [25]. The most notable and reliably measured difference between the two states is the increase in alpha power from EO to EC, as demonstrated by Barry et al. and Hartoyo et al. [24,29]. The results of the current study show consistent findings, indicating that the novel textile-based EEG is able to capture the defining characteristics of EEG-based differences between EO and EC. In addition, we examined the delta, theta, and beta power bands. These have been shown in prior work to follow a similar increase from EO to EC; however, the opposite pattern was noted in the current work. Unlike in the alpha band, where the change has been widely observed to occur across the cortex, theta and beta changes were reported to be more localized to posterior regions, and delta to frontal and lateral regions [24,30,31]. These topographical changes in delta, theta, and beta have been noted in studies utilizing whole-head EEG, but the current study employed only minimal frontally located recording channels. Localized activity opposite the measured region, along with differences in reference locations, may have resulted in inverse activity. Although we acknowledge the highly speculative nature of this explanation, we want to highlight that the power spectra results reported in prior works with the same data have reported similar trends in delta and theta activity [26]. Regardless, changes in frequency band powers are directly related to cognitive processes and provide a physiologically relevant context for a pBCI system. Delta and theta changes have been associated with increased attention during the performance of cognitive tasks and activation of the anterior cingulate cortex [32,33]. Changes in beta have been related to cognitive and emotional processing [34]. Alpha activity, the most widely studied, is linked to inhibition and is inversely related to mental processing. During EC, higher alpha power is indicative of the suppression of visual–spatial processing, freeing cognitive resources otherwise employed in visual processing [24,31,34,35,36].

The power band changes observed in the textile system were reflected in the dry-electrode EEG, which served as the baseline comparison. The results indicate that even with minimal electrodes, textile-based EEG can capture changes in cognitive states. Additionally, textile EEG captured notably smaller differences in the average power across bands than dry EEG. Lopez-Larraz et al. noted that textile EEG was more susceptible to artifacts; this was observed as significantly higher power in the 45–55 Hz range [26]. The structure of the textile electrodes or moisture absorbed by the fabric may have caused the increased signal artifacts. Moisture absorption is a potential drawback of this technology: sweating induced by physical activity or stress can be absorbed by the fabric and could cause shorts across the electrodes, rendering the signal unusable. However, as with traditional EEG electrodes, proper preparation leads to good signal quality. Steps can be taken to avoid excess moisture absorption and retain the practicality of a fully textile-based system.

Consistent with the power band analysis, the ML classification results revealed a similar performance (94% accuracy). Scores above 80% in all four metrics indicate strong performance in the classification of cognitive states. The performance of portable EEG for cognitive state assessment is further supported by [37], who demonstrated that a minimal wireless EEG system could successfully differentiate workload conditions and age-related effects during complex flight tasks, validating the utility of non-traditional approaches such as textile EEG. The accuracies for EO vs. EC classification in the textile (94%), dry (94%), and combined (92%) models are consistent with prior works, where SVM classification has been reported with accuracies ranging from 70% to 98% [25,38,39,40,41]. The features used for the classifier input consisted of both physiologically relevant (delta, theta, alpha, and beta mean spectral power) and statistically derived characteristics of the EEG signal. Gopan et al. examined EO vs. EC classification performance using only statistically derived features with SVM and KNN classifiers. The results found that skewness, kurtosis, and entropy had individual feature accuracies of 72%, 76%, and 68%, respectively [38]. The Gopan et al. study reveals that classification features extracted through purely statistical methods can be a supplement to the standard physiologically relevant EEG features. The generalizability assessment, which used the dry EEG data to train the model and textile EEG data for testing, resulted in comparable performance to either of the standalone models. This indicates that a pBCI system utilizing minimal electrodes is robust and adaptable to non-traditional textile-based electrodes. The accurate classification of EO and EC cognitive states on a platform well-suited to applications outside the laboratory is an important step towards the practical use of pBCIs.

Novel signal processing techniques provide additional support for the use of minimal electrode systems. Prior work from our group has shown that even single-channel EEG data can be successfully denoised. Using an empirical mode decomposition template-matching (EMD-TM) technique, ERPs were extracted with high correlation to the gold-standard ICA method (r = 0.88 for N100, r = 0.78 for P300, and r = 0.80 for N400 ERPs, all *p* < 0.05) [42]. Advanced time–frequency (TF) filtering techniques have also been developed that can effectively separate neural signals from ocular artifacts without requiring large electrode arrays, with comparable performance to ICA (*p* < 0.001) [43]. Methods like EMD-TM and the novel TF approach, which are specifically designed for out-of-laboratory settings on low-density systems, can be applied to textile-based EEG to further increase the effectiveness of pBCI systems. Additionally, electrode signals can be projected to more distant channels to predict activity by leveraging the principle of volume conduction. As demonstrated by [44], distant electrodes can be combined to predict Cz electrode signals with high accuracy, with both group-level and individual-level reliability. An SVM classifier applied to channel projections on an auditory oddball paradigm achieved 93.93% classification accuracy for tonal stimuli and 74.24% accuracy for word stimuli. This technique might be applied to project forehead EEG channels to regions relevant for cognitive state classification, thereby increasing the number of “sources” available to a pBCI system. In addition to novel denoising approaches, other work from our group has demonstrated the potential development of new pBCI solutions using novel measures. For example, we have shown that non-traditional textile electrodes can capture novel neurophysiological measures known as blink-related oscillations (BROs) [45]. BROs have been shown to index environmental monitoring processes involving multiple brain regions and have been shown to capture brain function changes in clinical conditions such as concussion and aging as well as in non-clinical situations such as aircraft pilot monitoring and working-memory evaluation in everyday settings [37,46,47,48].

The current study has several limitations, which might be addressed in future works. First, the sample size of 10 participants generated a minimum amount of data with which to train an ML classifier. This presents a major limitation of this work for model generalizability. Future work would utilize increased numbers of participants to validate the classification results from the current study and overcome the generalizability limitations. Second, the number of recording channels was limited by the surface area of the device [49]. In order to increase the recording density while retaining channel integrity, future work might incorporate light-based sensing (e.g., fNIRS) to maximize use of the available space without impacting EEG recording quality. Finally, the cognitive state being classified, EO vs. EC, is a reliable baseline measure, but in order for it to be adopted into everyday use, further validation of the system using more complex cognitive states is required. Future works would measure states such as fatigue, stress, mental workload, and vigilance, whose classification would allow a pBCI system to provide alerts and modulate the environment to prevent potentially life-threatening situations for those employed in fields such as aviation, space, and the military.

## 5. Conclusions

This study presents the first use of fully textile-based EEG for a pBCI application. The results showed that textile-based EEG was able to reliably capture the expected changes in EEG power bands between EO and EC, specifically the increase in alpha power, and that a pBCI system utilizing non-traditional textile electrodes is both accurate and has strong potential for generalizability. The successful use of textile-based EEG for pBCIs paves the way for applications in settings outside the laboratory where traditional EEG is incompatible. Future work would apply textile-EEG-based pBCIs to more complex cognitive states, with the goal of improving human–machine interactions in both clinical and non-clinical settings.

## Figures and Tables

**Figure 1 sensors-25-06080-f001:**
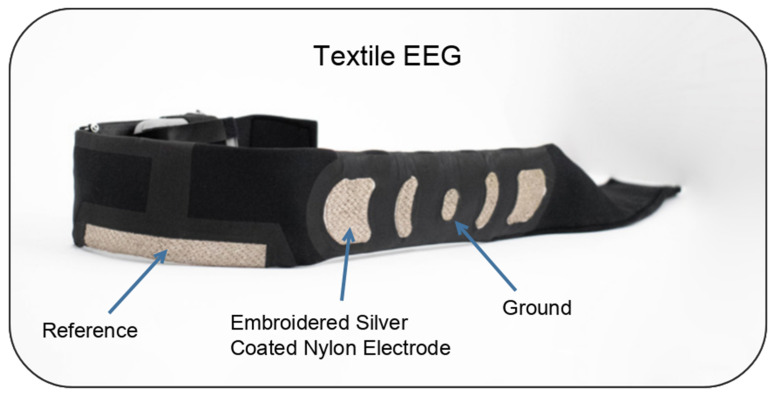
Textile EEG headband. Modified from Lopez-Larraz et al. [26].

**Figure 2 sensors-25-06080-f002:**
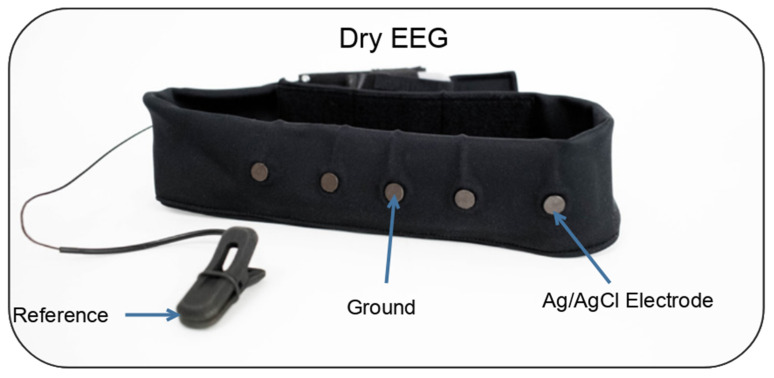
Dry EEG headband. Modified from Lopez-Larraz et al. [26].

**Figure 3 sensors-25-06080-f003:**
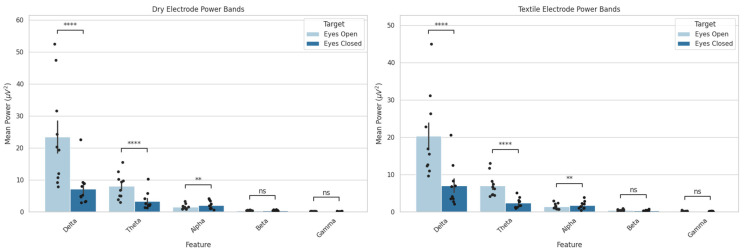
Dry and textile EEG mean power bands compared between eyes-open and eyes-closed conditions (ns: no significance; ** *p* < 0.01; **** *p* < 0.0001). Data are presented as mean ± standard deviation; dots represent individual participant data.

**Figure 4 sensors-25-06080-f004:**
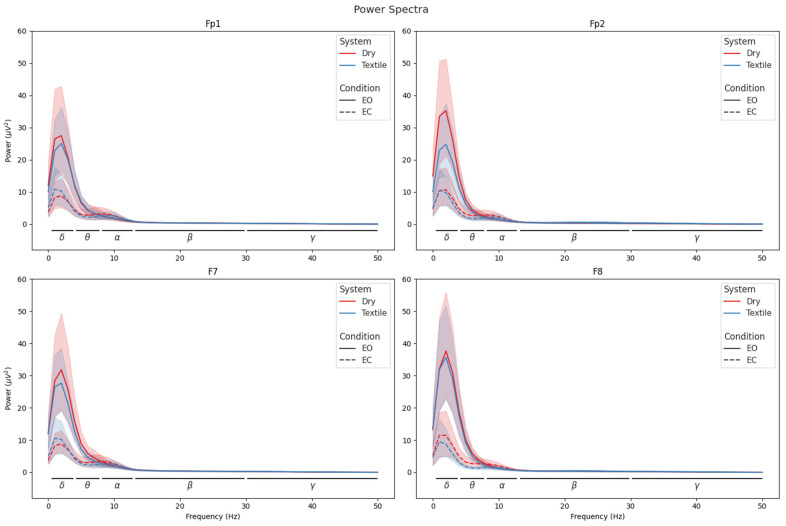
Subject mean power spectra at Fp1 and F7 (**left**) and Fp2 and F8 (**right**) for both EEG systems. Solid lines indicate eyes open (EO); dashed lines indicate eyes closed (EC). Solid black lines indicate delta (*δ*), theta (*θ*), alpha (*α*), beta (*β*), and gamma (*γ*). Shaded areas indicate confidence intervals.

**Figure 5 sensors-25-06080-f005:**
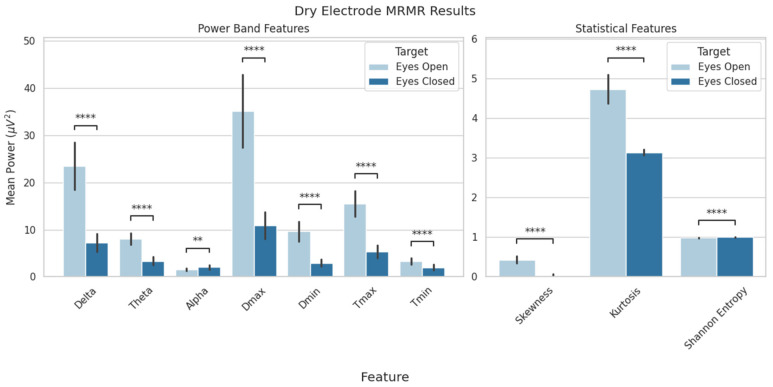
MRMR feature selection results for dry-electrode data. Features were compared between the eyes-open and eyes-closed conditions. Shown above are delta, theta, and alpha average power values; minimum and maximum delta and theta power; and skewness, kurtosis, and Shannon entropy (** *p* < 0.01; **** *p* < 0.0001). Data are presented as mean ± standard deviation.

**Figure 6 sensors-25-06080-f006:**
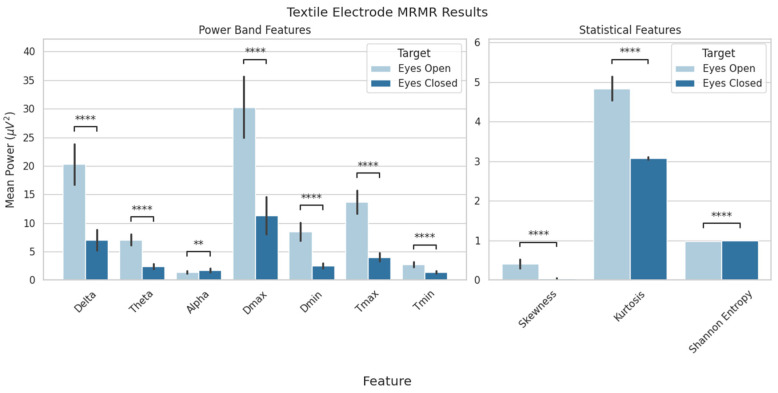
MRMR feature selection for textile-electrode data. Features were compared between the eyes-open and eyes-closed conditions. Shown above are delta, theta, and alpha average power values; minimum and maximum delta and theta power; and skewness, kurtosis, and Shannon entropy (** *p* < 0.01; **** *p* < 0.0001). Data are presented as mean ± standard deviation.

**Table 1 sensors-25-06080-t001:** Initial model selection cross-validation scores (mean ± standard deviation). Classifiers shown are Support Vector Machine with linear kernel (lSVM), Support Vector Machine with Gaussian kernel (gSVM), K-Nearest Neighbor (KNN), Random Forest (RF), and Linear Regression (LR).

Model Selection Results					
Model	Metric				
	Accuracy	Precision	Recall	F1	Fit Time
Dry					
lSVM	0.91 ± 0.10	0.90 ± 0.13	0.95 ± 0.11	0.92 ± 0.09	0.21 ± 0.10 ms
gSVM	0.75 ± 0.08	0.78 ± 0.16	0.78 ± 0.18	0.75 ± 0.09	0.08 ± 0.02 ms
KNN	0.88 ± 0.10	0.95 ± 0.12	0.82 ± 0.21	0.86 ± 0.12	0.06 ± 0.01 ms
RF	0.96 ± 0.08	0.97 ± 0.11	0.97 ± 0.08	0.97 ± 0.07	5.29 ± 0.09 ms
LR	0.88 ± 0.08	0.88 ± 0.13	0.90 ± 0.13	0.88 ± 0.08	0.42 ± 0.05 ms
Textile					
lSVM	0.95 ± 0.09	0.95 ± 0.12	0.97 ± 0.08	0.95 ± 0.08	0.14 ± 0.02 ms
gSVM	0.84 ± 0.16	0.86 ± 0.15	0.80 ± 0.20	0.82 ± 0.17	0.08 ± 0.01 ms
KNN	0.89 ± 0.11	0.89 ± 0.12	0.90 ± 0.13	0.89 ± 0.11	0.08 ± 0.01 ms
RF	0.97 ± 0.05	0.98 ± 0.06	0.97 ± 0.08	0.97 ± 0.05	5.21 ± 0.08 ms
LR	0.94 ± 0.09	0.93 ± 0.12	0.97 ± 0.08	0.94 ± 0.08	0.40 ± 0.06 ms
Dry/Textile					
lSVM	0.95 ± 0.02	0.90 ± 0.03	1.00 ± 0.00	0.95 ± 0.01	0.22 ± 0.09 ms
gSVM	0.76 ± 0.04	0.73 ± 0.07	0.82 ± 0.03	0.77 ± 0.03	0.08 ± 0.01 ms
KNN	0.82 ± 0.03	0.89 ± 0.04	0.74 ± 0.03	0.81 ± 0.03	0.06 ± 0.00 ms
RF	0.94 ± 0.02	0.92 ± 0.03	0.96 ± 0.02	0.94 ± 0.01	5.38 ± 0.25 ms
LR	0.91 ± 0.01	0.85 ± 0.01	1.00 ± 0.00	0.92 ± 0.01	0.44 ± 0.06 ms

**Table 2 sensors-25-06080-t002:** Cross-validation results (mean ± standard deviation) using linear kernel SVM. **Bold** numbers indicate significant (*p* < 0.05) findings compared with chance level.

Classification Results					
Model	Scoring Metrics				
	Accuracy	Precision	Recall	F1	Fit Time
Dry	**0.91 ± 0.10**	0.90 ± 0.13	0.95 ± 0.11	**0.92 ± 0.09**	0.21 ± 0.10 ms
Textile	**0.75 ± 0.08**	0.78 ± 0.16	0.78 ± 0.18	**0.75 ± 0.09**	0.08 ± 0.02 ms
Dry/Textile	**0.88 ± 0.08**	**0.88 ± 0.13**	**0.90 ± 0.13**	**0.88 ± 0.08**	0.42 ± 0.05 ms

## Data Availability

Original data for this study are available in [26].

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
