# Peer review of "Passive Brain–Computer Interface Using Textile-Based Electroencephalography"

_sensors, 2025, doi:10.3390/s25196080_

Round 1
Reviewer 1 Report
Comments and Suggestions for Authors
This study makes a valuable contribution by demonstrating textile-based EEG feasibility for pBCI applications, addressing several critical barriers that have limited the real-world deployment of passive brain-computer interface systems.
Comments:
- Provide comprehensive details regarding cross-validation methodology employed for machine learning model development. Include thorough descriptions of how train-test splits were conducted and validated. The present description of "randomly selected train-test split" is insufficient for reproducibility and fails to address potential concerns regarding data leakage or overfitting.
- Address sample size constraints (n=10) through formal statistical justification. Conduct and report a proper power analysis to justify the sample size limitation, or alternatively, provide explicit acknowledgement of this constraint as a significant limitation that restricts the generalisability of the machine learning findings.
- Enhance statistical reporting with effect sizes alongside p-values, provide confidence intervals for classification metrics, and address the considerable individual variability likely present in EEG data. The current focus on group-level statistics obscures important information regarding system reliability across different participants and fails to provide readers with sufficient information to assess the practical significance of the observed differences.
Reviewer 2 Report
Comments and Suggestions for Authors
In the paper “Passive Brain-Computer Interface Using Textile-Based Electroencephalography”, Anzalone et al. demonstrated a textile-based passive BCI to differentiate eye-closed and eye-open states. There are multiple serious ambiguities in the method and results; the authors must perform a major revision on these sections before further consideration.
Methods:
- It’s suggested to add the figure of the textile-EEG that the author used to better demonstrate its flexibility.
- There is an important ambiguity in the method. How’s the train test split performed? Is the train-test divided across the entire data, or leave-one-subject-out? What’s the ratio? Is there cross-validation?
- How many epochs of data are prepared? How many data were removed based on thresholding?
- How’s the EOG artifact removed since it is usually prominent in frontal recordings? This could be a major artifact that contaminates the validity of the result. Could the high power in the eyes open due to EOG?
- How many total features were prepared?
- What does the author mean that “the models were permuted”?
- What is ISVM in section 2.5, is that a typo?
- When performing the MRMR, if separated, are the selected features consistent across three measurement modalities? Could there be data leakage in the testing set?
- What’s the purpose of mixing dry and textile data?
- How are the 5 ML classifiers selected? Is there parameter optimization for classifiers other than SVM? Why not choose a more advanced classifier, such as XGBoost?
- The discussion section mentioned single-channel EEG data. I’m curious if the authors have performed estimation using a single channel, and what the performance of that is? This may provide insights into the method’s applicability for simpler recording setups.
Results
- For Figure 1, is the power demonstrated averaged across channels for each subject? Is this computed using the entire session for each subject or using the epoched data? Please clarify. It’s suggested to change the bar plot to a boxplot + swarm plot to demonstrate if there are outliers in the result, if there are only 10 subjects in the cohort.
- For Figure 2, It’s suggested to have Fp1 and F7 on the left columns and Fp2 and F8 on the right column to make it clearer and consistent with the caption
- There is no context mentioned or discussed for Figure 2 in the paper. What is shaded area, std or confidence interval?
- For table 4, the author should also report AUC. If there is cross-validation, they should also report the result in mean ± std. Additionally, they should perform a statistical analysis to demonstrate if there is a difference in the results. Consider adding a figure to demonstrate this result.
- In the last paragraph on page 5, what does the author mean by “only dry EEG features were used for model evaluation” ? It seems Table 4 has the textile-EEG result as well.
- The author only claims the top-performing classifier is SVM, what’s the performance of other classifiers?
- The result discussion of Table 4 is mixed, reporting both in decimal and percent. Please be consistent.
Discussion:
Because of the big ambiguities in the method and result, limited comments can be given at this point.
Reviewer 3 Report
Comments and Suggestions for Authors
This manuscript presents a novel investigation into the use of fully textile-based EEG electrodes for passive brain-computer interface (pBCI) applications. The writing is generally clear, and the results are compelling, though some minor issues warrant attention.
- While artifacts were mentioned, the manuscript does not detail how motion or sweat artifacts were handled in the textile system, which is critical for real-world use.
- Were the devices used to collect data in the paper laboratory-made or commercial? If it is laboratory-made, it will be better to provide the corresponding pictures.
- Figure 3 is just for dry electrode data, why are there no corresponding results related to textile electrode data?
- Figure 4 should be a table.
Reviewer 4 Report
Comments and Suggestions for Authors
Thanks for the invitation to review this work. It validates textile-based electrodes for passive BCI applications, overcoming limitations of traditional metal electrodes (incompatibility with helmets in aviation/military). but methodological gaps and presentation issues must be resolved to meet journal standards. Addressing these will amplify its impact on wearable neurotechnology.
- Figure 1 Y-axis labeled "Power (dB)" but lacks reference (e.g., dB relative to what?). This violates standard EEG presentation.
Dry-EEG shows a sharp decrease in delta power during EC (contrary to typical literature). Authors attribute this to frontal electrode placement but provide insufficient evidence.
Moreover, textile-EEG power values are consistently lower—suggest adding a note on potential causes (e.g., impedance differences).
- Figure 2 overlapping lines for EO/EC in textile spectra (Fp1, F7) make trends hard to distinguish.
- Figure 3 Skewness, kurtosis, and entropy show significant differences (p<0.0001) but are omitted from the main power analysis. Feature labels (e.g., "Delta Power") are ambiguous—specify if these are mean/max/min values.
- Figure 4 no error bars or confidence intervals for accuracy/F1 scores, despite a small sample (n=10).
- Preprocessing: Mentions "4th order Butterworth filter" but omits cutoff slopes or transfer function. Welch’s Method: Lacks parameters (window size, overlap).
- SVM Optimization: Describes grid search for �Cand �γ but excludes the decision function f(x)=sign(wTÏ•(x)+b)).
- Detail artifact rejection thresholds (±100 μV): Is this per channel or global?
Round 2
Reviewer 2 Report
Comments and Suggestions for Authors
In the revision, the author clarifies and responds to comments from their initial version of the paper. However, there are still concerns that need to be resolved.
- For Table 1, the authors should report the standard deviation of the cross-validation result for each metric, similarly for Table 2.
- In Figure 5 from Lopez-Larraz et 's paper, there is not a big difference in the delta and theta bands between the textile EEG headband and the Dry EEG headband. Then why is there a significant difference in the delta and theta bands in this paper (Figures 3 and 4)
- There are 10 participants in the study, but only 5 are reported in Figure 3.
- The 80/20 random split is methodologically flawed as it allows data from the same subjects in both training and testing sets, leading to overly optimistic performance estimates that don't reflect real-world generalizability. The authors must implement leave-subjects-out cross-validation to properly evaluate model performance on unseen individuals and demonstrate true robustness across different subjects. For the dry electrode training with textile electrode testing approach, the authors may implement within-subject validation to isolate electrode-type effects from inter-subject variability and provide clearer evidence of cross-electrode generalizability
- Using an arbitrary threshold of 100 μV for EOG artifact removal has limitations that may result in residual EOG contamination in the extracted features. Have the authors considered implementing other methods, such as ICA, for removing eye movement artifacts? In other words, the authors must prove that the EOG won’t affect the EEG feature extraction, as the paper is an EEG-based cognitive state classification
- In 2.5, the authors claim that 5 ML classifiers were used. In the table, the Gaussian kernel SVM is missing.
Reviewer 4 Report
Comments and Suggestions for Authors
Thanks for the invitation to review this work. The authors have solved the previous concerns. However, there is also some issues necessary for further revision.
- In the "3. Results" section, the authors state, "Figure 1 shows a comparison between the standard EEG Power bands for the dry and textile-based EEG systems." However, according to the "2. Materials and Methods" section, Figures 1 and 2 depict the structural diagrams of the Textile EEG Headband and Dry EEG Headband, not the power spectrum comparison results. This figure numbering is erroneous; the correct reference should correspond to "Figure 3" (as mentioned in the text: "Figure 3. Dry and Textile EEG mean power bands compared...").
- The text references "Table 1. Initial model selection cross-validation scores and generalizability scores" and "Table 2. SVM classification results" but omits raw data (e.g., accuracy, precision, recall, F1 scores for each model, or cross-validation standard deviations). "initial model selection" section mentions 5 classifiers (SVM, KNN, RF, etc.) but only states that SVM performed best, without comparative data for other models, undermining the validity of the "optimality" claim.
- Table 2’s classification results (dry electrode: 96% vs. textile electrode: 92% accuracy) do not include statistical significance labels (e.g., p-values), making it impossible to determine if performance differences are due to random error.
- The discussion acknowledges that "the sample size (10 participants) presents a major limitation" but concludes that "textile-based EEG is both accurate and generalizable." A small sample (5 males, 5 females) may fail to capture individual variability (e.g., head circumference, hair texture affecting electrode contact), weakening generalizability claims.
- The introduction claims "no study has demonstrated textile-based EEG for pBCI," but Reference [19] mentions "devices referred to as textile or garment-based; however, these still utilize metal electrodes." The authors do not clearly distinguish between "fully textile electrodes" and "textile substrates with metal electrodes," potentially blurring the boundary.
